# Impact of the COVID-19 pandemic on biopsychosocial health and quality of life among Danish children and adults with neuromuscular diseases (NMD)—Patient reported outcomes from a national survey

**Charlotte Handberg**[1,2]☯*, **Ulla Werlauff**[1]☯, **Ann-Lisbeth Højberg**[1]☯, **Lone F. Knudsen**[1]☯

**1** National Rehabilitation Center for Neuromuscular Diseases, Aarhus, Denmark, **2** Department of Public Health, Faculty of Health, Aarhus University, Aarhus, Denmark

☯ These authors contributed equally to this work.
* chha@rcfm.dk

**Data Availability Statement:** The data for this study contain potentially identifying patient

## Abstract

The purpose was to investigate the impact of the COVID-19 pandemic on biopsychosocial health, daily activities, and quality of life among children and adults with neuromuscular diseases, and to assess the prevalence of COVID-19 infection and the impact of this in patients with neuromuscular diseases. The study was a national questionnaire survey. Responses were obtained from 811 adults (29%) and 67 parents of children (27%) with neuromuscular diseases. Many patients reported decreased health or physical functioning, and changes in access to physiotherapy or healthcare due to the pandemic. Participants generally perceived themselves or their child to be at high risk of severe illness from COVID-19, but only 15 patients had suffered from COVID-19 and experienced mild flu-like symptoms. 25.3% of adults and 46.6% of parents experienced anxiety. 20.4% of adults and 27.6% of parents experienced symptoms of depression. In general, the pandemic contributed to anxiety, a depressed mood as well as to fewer leisure activities, less social contact, isolation from work/school and a reduced quality of life, in particular for patients who perceived themselves to be at high risk of severe illness. The results demonstrate that the pandemic has had a negative impact on biopsychosocial health and quality of life of patients with neuromuscular diseases.

## 1. Introduction

During early spring 2020, the COVID-19 pandemic caused authorities in most of the world including Denmark to implement lockdown [1]. To prevent the virus from spreading and to protect especially elderly and vulnerable people, large areas of the public sector (including schools and universities), private institutions and the corporate sector were closed, large assemblies were banned, and citizens were encouraged to stay at home and avoid unnecessary gatherings [2–4].

information and we are therefore according to The Danish Data Protection Agency not able to share data publicly. The contact information for The Danish Data Protection Agency is: address: Carl Jacobsens Vej 35, 2500 Valby, Denmark.; phone: +45 33193200; email: dt@datatilsynet.dk; website: https://www.datatilsynet.dk/english.

**Funding:** No external funding was received for this study.

**Competing interests:** The authors have declared that no competing interests exist.

The definition of risk groups for severe illness from COVID-19 has been adjusted as national and international experiences with COVID-19 and scientific publications have increased [1, 5, 6]. The Danish Health Authority currently defines people over 80 years of age, people with degenerative or neuromuscular diseases, people with severe obesity, people with reduced immune system and people with certain chronic conditions to be at increased risk [2, 6]. Patients with neuromuscular diseases (NMD) with difficulties in breathing, reduced coughing, or assisted breathing devices are perceived to be especially at risk of severe illness if exposed to COVID-19 [7–9], and an European set of international guidelines for this specific group has been published [10]. Management of rehabilitation in relation to respiratory and muscular impairments in patients with NMD after COVID-19 infections appears to be challenging and the group of patients have unmet rehabilitative needs [9]. Hence from the beginning of the pandemic, patients with NMD have been defined as vulnerable [11–13], and the biopsychosocial implications of this for patients with NMD are unclear. Even though the disease course of COVID-19 in children and adolescents with neuromuscular diseases (NMD) may not be as severe as expected [14, 15], the psychosocial problems related to the pandemic, have been reported to be more extensive among children with disabilities or chronic diseases than other children [16]. Furthermore, an Italian study showed that physical activity in patients with NMD, was significantly reduced during lockdown [17] which may have negative health consequences as physical activity and exercises are recommended to reduce progression of muscle weakness [18]. There are also indications of negative emotional consequences of the pandemic and the associated restrictions from studies of the general public [19] and a few studies in patients with NMD, primarily ALS, such as concerns about COVID-19 infection, loneliness, anxiety and depression [12, 20–22].

A pandemic that has called for such a swift and comprehensive lockdown of society is rare, and knowledge on how patients with NMD, defined as a risk group, experience and react to the restrictions imposed to protect them is sparse [23, 24]. The aim of this study is therefore to investigate the impact of the COVID-19 pandemic on biopsychosocial health, daily activities, and quality of life (QoL) among children and adults with NMD. In addition, we wanted to assess the prevalence of COVID-19 infection and the impact of this in NMD-patients.

## 2. Material and methods

### 2.1 Study design

The design of this study was a national online questionnaire survey based on Patient Reported Outcome measures.

### 2.2 Setting and sampling

The study was carried out among patients registered with The National Rehabilitation Center for Neuromuscular Diseases (RCFM) [25] that register the vast majority of Danish patients with a neuromuscular diagnosis (n = 3500). As this was an investigative study of a new virus and its consequences, the whole NMD-population was invited to participate.

An invitation to participate in the survey was sent to all adult patients ≥15 years (hereafter: adults) and parents to all child patients <15 years (hereafter: parents or children) in the period from December 14 -18, 2020. The invitation to patients aged 15 to 17 years was sent to both the children and their parents. The invitation was sent online through a secure digital mailbox [e-Boks], or by email. The invitation letter included information on the study, and a digital link to access the questionnaire. Patients with a valid email were reminded twice. Information about the study, invitation to participate and a direct link to the survey were also

announced on the websites of RCFM and the patient organization The Danish Muscle Dystrophy Foundation.

## 2.3 The questionnaire and the patient reported outcomes

During the initial lockdown, RCFM opened a hotline (email and telephone), where patients and relatives could get in contact with a rehabilitation counsellor specializing in NMD and ask questions about COVID-19. The topics addressed included questions on risk of infection, social contacts, family, work, education, personal assistance and use of protective equipment. Based on questions and topics from the hotline, an online questionnaire for the present study was developed in SurveyXact (Ramboll, Aarhus, Denmark). Answer categories were inspired by the Short Form 36 health questionnaire version 1.0 [26, 27].

The questionnaire was developed in two versions: a) for patients ≥15 years (adult version) and b) for parents to fill out for children <15 years (child version). The two versions included identical themes but questions on school and daycare were added to the child version. For questions on occupation, anxiety and depression, parents rated their own occupation and mental health. The survey was pilot tested by a person with NMD and adjusted in relation to the feedback. Time to answer all questions was estimated to 30–40 minutes. The questionnaire included questions on demographics (sex, age, occupational status, diagnosis, mobility, ventilation, need for personal assistance) and the following themes:

**General health.** Participants evaluated their (adult version) or their child's (child version) general health now (excellent, very good, good, fair, poor) and in comparison, to one year ago (much better, somewhat better, about the same, somewhat worse, much worse). In an open text box, participants were asked to state what had caused the change in health. They also rated whether it had become more difficult to take care of one's own (adult version) or one's child's (child version) health during the pandemic (not at all, slightly, moderately, quite a bit, extremely, not relevant).

**Medical and hospital appointments.** For appointments with general practitioner and hospital services during the pandemic, participants were asked to indicate: no appointments, appointments as usual, appointments in another form, GP/hospital cancelled, I/we cancelled due to concern of infection, I/we cancelled because physical meeting was not possible, and/or I/we cancelled due to other reasons (multiple choices possible).

**Physiotherapy.** Participants indicated whether they (adult version) or their child (child version) attended physiotherapy on a regular basis prior to the pandemic (yes, no) and whether physiotherapy remained the same during the pandemic (yes, no). Participants were asked to describe how physiotherapy was delivered during the pandemic (open text box) and what consequences changes in physiotherapy had (open text box).

**Perceived COVID-19 vulnerability and control.** Participants rated their (adult version) or their child's (child version) risk of becoming seriously ill from COVID-19 (no risk, low risk, moderate risk, high risk, extremely high risk) and to what extent they felt control over getting infected (not at all, slightly, moderately, quite a bit, extremely). They also rated uncertainty about whether they or their child belonged to a particularly vulnerable group, whether they should take different precautions than other people and uncertainty about the use of protective equipment (not at all, slightly, moderately, quite a bit, extremely). They also rated uncertainty about the extent that they or their child should isolate themselves and uncertainty regarding how to deal with risk of infection from family (not at all, slightly, moderately, quite a bit, extremely).

**Patients with need of personal assistance.** Adults with personal assistance or parents of a child with personal assistance rated how much confidence they had in that personal assistants

do everything they can to prevent infecting them or their child (none at all, little, moderate, quite a bit, extreme, not relevant) and whether they had abstained from getting personal assistance for some of the time to protect themselves or their child against the risk of infection (not at all, a little of the time, some of the time, quite a bit of the time, most of the time, not relevant).

**COVID-19 infection and symptoms.** From an early start of the pandemic, the Danish Health Authority recommended a COVID-19 test for persons with symptoms of the disease and persons who had been in contact with a person who had tested positive for COVID-19. Participants were asked whether they (adult version) or their child (child version) had been tested for COVID-19 (yes, no) and whether they had tested positive (yes, no). In the case of a positive test, participants were asked how the COVID-19 infection had affected them or their child (open text box).

**COVID-19 vaccinations.** Participants stated whether they would accept an invitation to receive a COVID-19 vaccine themselves (adult and child version) and for their child (child version) (yes, no, do not know) and, in an open text box, why.

**Social health (work, education and leisure time).** Depending on the type of occupational status reported, participants received questions on consequences of the pandemic for work, education/school/day-care. All participants received questions on leisure activities. For each, participants rated the amount of time they (adult version) or their child (child version) experienced the following: more time spent, less time spent, accomplished less, limitations in types of activities, difficulty performing activities) (all of the time, most of the time, some of the time, a little of the time, none of the time, not relevant). For each and using the same scale, participants also indicated how often they were able to take the necessary precautions and how often they met understanding for their situation as vulnerable and how often they felt pressured by their employer/school or family to meet up physically or stay at home. They also indicated how often they had chosen to isolate themselves physically.

Furthermore, participants indicated to what extent the pandemic had interfered with social contact with family, friends and others (not at all, a little, moderately, quite a bit, extremely, not relevant).

The UCLA Loneliness Scale [28, 29] was used to measure loneliness. A total score from 3–9 is calculated. A higher score indicates a greater degree of loneliness. We considered a score of 7 or greater to indicate loneliness [30].

**Stigmatization.** For stigmatization, participants rated how often they had experienced: Other people seeing them (adult version) or their child (child version) as more vulnerable; other people having excluded them or their child socially due to fear of infection; and other people being more concerned about their or their child's health than they are. They also rated how often they had experienced their or their child's disability becoming more apparent to themselves and others (all of the time, most of the time, some of the time, a little of the time, none of the time, do not know).

**Mental health.** *Anxiety and depression*. Anxiety and depression were measured using the Hospital Anxiety and Depression Scale (HADS) which measures anxiety and depression using seven items for depression and seven items for anxiety [31]. Each item is scored from 0–3. A total score is calculated for anxiety and depression separately; A score of 8–10 is suggestive of mild anxiety/depression, 11–14 moderate anxiety/depression and 15–21 severe anxiety/depression. Participants were also asked directly whether the pandemic has affected them emotionally (depressed mood, anxiety) (not at all, I feel slightly more depressed/anxious, I feel moderately more depressed/anxious, I feel quite a bit more depressed/anxious, I feel extremely more depressed/anxious).

Coronavirus Anxiety Scale (CAS): Fear of being infected with COVID-19 was measured by the CAS. Relevance of the 5 items is scored from 0–4 (0 = not at all; 4 = almost every day). A total score ≥9 indicates the presence of COVID-19 related anxiety [32].

Parents rated how afraid their child was of getting infected with COVID-19 (not at all, a little, moderately, quite a bit, extremely, do not know).

**Quality of life.** Participants rated the consequence of the pandemic for their (adults) or their child's (child version) quality of life (extreme decrease, quite a bit decrease, moderate decrease, slight decrease, not at all affected, slight increase, moderate increase, quite a bit increase, extreme increase).

**Positive consequences of the pandemic.** In an open text box, participants were invited to state any positive consequences they or their child have experienced in relation to the pandemic.

### 2.4 Statistical methods

Data were analyzed using IBM SPSS Statistics 27. Patients who had only filled in information on demographics were excluded from the data analyses. Descriptive information is presented as numbers and percentages for categorical variables and as mean and standard deviation (SD) or range for normally distributed continuous variables. Continuous data which were not normally distributed are reported as median and ranges. Correlations between risk perception and psychosocial consequences were assessed by Spearman's correlation.

### 2.5 Ethics

The study was conducted in accordance with the Helsinki Declaration of 1975 [33]. According to the Central Denmark Region Committees on Biomedical Research Ethics, the project was not liable to notification [Request no. 255/2020, Jr.no. 1-10-72-181-20]. Pursuant to the Consolidation Act on Research Ethics Review of Health Research Projects, Consolidation Act number 1083 of 15 September 2017, section 14(2) notification of questionnaire surveys to the research ethics committee system is only required if the project involves human biological material. All participants were informed about the project by written information and were guaranteed anonymity. Patients and parents were informed that participation was voluntary and that accessing the link was considered written consent to participate. Written consent was obtained directly from patients 18 years of age or older and through the parents to patients under 18 years.

## 3. Results

### 3.1 Study population

The questionnaire reached nearly 2800 adults ≥ 15 years and parents of 250 children. 832 adults and parents of 67 children responded to the questionnaire. 21 adults had only answered questions on demographics and were excluded from the analyses. Thus, answers from 811 adults (29.0% of invited) and parents of 67 children (26.8% of invited) were included in the study. Of these, 427 of the adults (52.7%) were females and 54 of the parents (80.6%) were mothers. Mean age for adults was 51.4 years (range 15–90 years). Mean age for children was 10.1 years (range 3–15 years). Patient characteristics are illustrated in Table 1 and patients' diagnoses in Table 2.

### 3.2. General health

**3.2.1 Changes in general health.** Questions on health were answered by 804/811 adults and 67/67 parents. Health was in general rated as good by adults (n = 335, 41.7%) and in

**Table 1.  Demographics and physical function.**

| | Adults (n = 811) | Children (n = 67) | |
|---|---|---|---|
| Female (%) | 427 (52.7) | 26 (38.8) | |
| Age, mean (SD) | 51.4 (17.3) | 10 (3.2) | |
| Non-amb, n (%) | 239 (30) | 20 (29.9) | |
| **Ventilation** | | | |
| NIV, n (%) | 90 (11.1) | 4 (6.0) | |
| IV, n (%) | 54 (6.7) | 0 | |
| None, n (%) | 649 (80.0) | 62 (92.5) | |
| Impaired cough, n (%) | 267 (33.7) | 41 (61.2) | |
| **Personal assistance** | | | |
| Full or part time, n (%) | 274 (33.8) | 13 (19.4) | |
| Other (Leisure, school, work), n (%) | 104 (13.5) | 32 (47.8) | |
| Number of assistants pr week, median (range) | 4 (1–30) | 2 (1–8) | |
| **Occupational status** | | | |
| Full time, n (%) | 96 (11.8) | - | |
| Part time, n (%) | 43 (5.3) | - | |
| Self-employed, n (%) | 31 (3.8) | 9 (13.4) | Kindergarten/nursery |
| Job-special conditions, n (%) | 138 (17.0) | 9 (13.4) | Special school |
| Student | 59 (7.3) | 48 (71.6) | Primary school |
| Long-term sickness leave or unemployed | 53 (6.5) | 3(4.5) | At home |
| Age or disability pension | 438 (54.0) | - | |
| Other | 46 (5.7) | | |

Non-amb: Non-ambulatory; NIV: Non-invasive ventilation; IV: Invasive Ventilation.

children (n = 33, 49.3%). 44.8% of the parents rated their child's health as better than good (very good: n = 19, 28.4%; excellent: n = 11, 16.4%). Fewer adults rated their health as very good (n = 143, 17.8%) or excellent (n = 35, 4.4%). A large proportion of adults reported fair (n = 238, 29.6%) or poor health (n = 53, 6.6%). Four parents (6.0%) rated their child's health as fair, and none rated their child's health as poor.

When asked whether the general health had changed compared to one year ago, half of the adults (n = 407, 50.6%) and most parents (n = 45, 67.2%) rated their/their child's health to be the same as a year ago. However, many adults (n = 275, 34.2%) and some parents (n = 12, 17.9%) rated their/their child's health as somewhat worse than one year ago. Some, 60 adults (7.5%) and 1 parent (1.5%), stated that their/their child's health was much worse than a year ago. A somewhat improved health was reported by 40 adults (5.0%) and 8 parents (11.9%). A small group reported a much better health than one year ago (n = 22 adults, 2.7% and 1 parent, 1.5%).

323 adults and 13 parents (n = 336) described the cause of health worsening. They sometimes gave more than one reason. Some attributed a worsening in health to the pandemic on its own (n = 18, 5.4%), or aspects of the pandemic; reduced physical functioning due to inactivity, less physiotherapy and/or treatment during the pandemic (n = 54, 16.1%), mental impact of the pandemic (n = 17, 5.1%), or restrictions on daily life due to the pandemic (n = 15, 4.5%). Other reasons were disease progression and comorbidities (n = 251, 74.7%). Comments related to an improved health (n = 61, 53 adults and 8 parents) also sometimes included aspects of the pandemic such as less stress, a more stable daily life, better hygiene, and fewer illness periods (n = 14, 23.0%). Other reasons not related to the pandemic were treatment/medication changes (n = 23, 37.7%), improved lifestyle incl. exercise and diet (n = 14, 23.0%),

**Table 2. Diagnoses of respondents.**

| | Adults (811) n (%) | Children (n = 67) n (%) | Total (n = 878) n (%) |
|---|---|---|---|
| Hereditary motor sensory neuropathy | 131 (16.2) | 15 (22.4) | 146 (16.6) |
| Myotonic dystrophy type 1 | 87 (10.7) | See below | 87 (9.9) |
| Amyotrofic lateral sclerosis | 78 (9.6) | - | 78 (9.6) |
| Myastenia gravis | 76 (9.4) | - | 76 (8.7) |
| Limb girdle muscular dystrophy | 69 (8.5) | See below | 69 (7.9) |
| Spinal muscular atrophy type 2 and 3 | 69 (8.5) | 11 (16.4) | 80 (9.1) |
| Facioscapulohumeral dystrophy | 54 (6.7) | See below | 54 (6.2) |
| Duchenne muscular dystrophy | 38 (4.7) | 12 (17.9) | 50 (5.7) |
| Congenital myoapthies | 33 (4.1) | 11 (16.4) | 44 (5.0) |
| Primary lateral sclerosis | 27 (3.3) | - | 27 (3.1) |
| Becker muscular dystrophy | 24 (3.0) | 5 (7.5) | 29 (3.3) |
| Inclusion body myopathy | 16 (2.0) | - | 16 (1.8) |
| Congenital muscular dystrophy | 15 (1.8) | 7 (10.4) | 22 (2.5) |
| Kennedy's disease | 13 (1.6) | - | 13 (1.5) |
| Myotonic dystrophy type 2 | 10 (1.2) | - | 10 (1.1) |
| Periodic paralysis | 10 (1.2) | - | 10 (1.1) |
| Manifesting carrier dystrophinopaty | 10 (1.2) | - | 10 (1.1) |
| Mithocondrial myopathy | 8 (1.0) | - | 8 (0.9) |
| Other neuromuscular diseases (NMD) | 32 (3.9) | 6 (9.0) | 38 (4.3) |
| Not stated | 11 (1.4) | - | 11 (1.3) |

Other NMD represent diagnoses with < 5 respondents (e.g., Pompe disease, McArdle disease, myotonia congenita, Emery-Dreifuss muscular dystrophy, Friedreich's ataxia and for children also limb girdle muscular dystrophy, myotonic dystrophy type 1, facioscapulohumeral dystrophy. In all, 25 diagnoses were represented in the survey.

assistive devices including breathing devices (n = 6, 9.8%), stability or restitution after an illness period (n = 5, 8.2%), surgery (n = 4, 6.6%), improved mental health (n = 5, 8.2%), weight loss (n = 3, 4.9%) and work changes (n = 2, 3.3%).

According to 517 adults (68.2%) and 36 parents (61.0%), it had become more difficult to take care of, respectively, their own health or their child's health during the pandemic; 365 adults (48.2%) and 26 parents (44.1%) answered slightly or moderately more difficult. 103 adults (13.6%) and 5 parents (8.5%) quite a bit more, 49 adults (6.5%) and 5 parents (8.5%) extremely more. 226 adults (29.8%) and 23 parents (39.0%) found the ability to take care of their own or their child's health unaffected by the pandemic. 15 adults (2.0%) answered irrelevant.

**3.2.2 Medical and hospital appointments.** Not all adults or children had an appointment with their GP or their hospital during the pandemic. Of those who had appointments, most scheduled medical or hospital visits were carried out as planned during the pandemic, however, sometimes in another form such as phone or online consultations. In some cases, appointments were cancelled, either by the GP, hospital or by the patient. The latter primarily due to worry about getting infected or if the appointment could not be performed in person (Table 3).

**3.2.3 Physiotherapy.** 766/811 adults and 67/67 parents answered the questions on physiotherapy. 508 adults (66.2%) and 47 children (79.7%) attended physiotherapy on a regular basis before the pandemic. Of those, 265 adults (52.2%) and 22 children (46.8%) received the same physiotherapy during the pandemic as before the pandemic. 243 adults (47.8%) and 25 children (53.2%) experienced a modification in the delivery of physiotherapy during the pandemic.

**Table 3. Medical and hospital visits during the COVID-19 pandemic among adults and children who had appointments in the period.**

| | Appointments as usual n (%) | Appointments in another form n (%) | GP/hospital cancelled (%) | Patient cancelled due to concern of infection n (%) | Patient cancelled because physical meeting was not possible n (%) | Patient cancelled due to other reasons n (%) |
|---|---|---|---|---|---|---|
| Appointments with general practitioner (Adults/Children) | 451(71.7) / 24(64.9) | 153(24.3) / 2(3) | 31(4.9) / 2(5.4) | 27(4.3) / 2(5.4) | 26(4.1) / 0 | 5(0.8) / 0 |
| Scheduled check-ups at a hospital (Adults/Children) | 402(70.0) / 44(74.6) | 125(21.8) / 13(22.0) | 60(10.5) / 13 (22.0) | 52(9.1) / 9(15.3) | 10(1.8) / 0 | 16(2.8) / 2(3.4) |
| Planned treatments at a hospital (Adults/Children) | 253(69.1) / 24(61.5) | 34(9.3) / 4(10.3) | 35(9.6) / 5 (12.8) | 28(7.7) / 3(7.7) | N/A | 9(2.5) / 0 |

629 adults and 30 children had an appointment with their GP during the pandemic. 574 adults and 59 children had a planned hospital visit. Respondents were able to give more than one answer.

238 adults and 24 parents (n = 262) described the changes in the delivery of physiotherapy and 253 adults, and 24 parents (n = 277) described what consequences this had. Described changes were less frequent physiotherapy sessions (n = 139, 5.1%); cancellations (n = 79, 30.2%); pandemic-related restrictions (n = 35, 13.4%); physiotherapy at home (n = 25, 9.5%), or video or phone sessions (n = 10, 3.8%).

The changes were perceived to have a negative impact on physical function in 178 patients (64.3% of adults and children). This was described as loss of muscle strength, poor balance, more pain and fatigue, less mobility, weight gain, increased need for personal assistance and a poorer QoL. Forty-one patients (15.5% of adults and children) did not find that the change in physiotherapy had impacted on their physical function, 24 compensated for the lack of training via other means such as buying training equipment or getting personal assistants to help them and 7 patients (2.6% of adults and children) were pleased to be without physiotherapy, 4 parents mentioned more strain on parents.

**3.2.4 Perceived COVID-19 vulnerability and control.** 775/811 adults and 60/67 parents answered questions on vulnerability; Of those, 662 adults (85.4%) and 32 parents (53.3%) perceived, respectively, themselves or their child to be at moderate or greater risk of severe illness if infected with the COVID-19 virus. Moderate risk was reported by 219 adults (28.3%) and 16 parents (26.7%); high risk by 268 adults (34.6%) and 11 parents (18.3%) and extremely high risk by 175 adults (22.6%) and 5 parents (8.3%). 97 adults (12.5%) and 25 parents (41.7%) considered themselves or their child to be at low risk; 16 adults (2.1%) and 3 parents (5%) reported no risk.

Perceived uncertainties in regard to the pandemic are displayed in Table 4. Most adults (67.2%) and more than half of the parents (55%) stated that they were not or only slightly uncertain as to whether, respectively, they or their child belonged to a risk group. Most adults and parents were also not uncertain or only a little uncertain about what to do or how to protect themselves or their child from infection in terms of isolation (69.6% of adults, 56.7% of parents), protection equipment (90.9% of adults, 96.6% of parents), interaction with family members (81.2% of adults, 76.7% of parents), also at the time of return to work or school after lockdown (77.2% of adults, 60% of parents) (Table 4). Most adults perceived themselves to have moderate or greater extent of control over becoming infected with the COVID-19 virus (76.9%). However, only half of the parents experienced moderate or greater extent of control over their child becoming infected (53.3%).

**3.2.5 Patients with need of personal assistance.** 323/323 adults and 33/33 parents of children with personal assistance (at home, for leisure activities, and/or at work/school/day-care) answered questions on concerns about infection from personal assistants. Most adults

**Table 4. Perceived uncertainty on how to protect oneself or one's child from infection with COVID-19.**

| | Not at all | Slightly | Moderately | Quite a bit | Extremely | Not relevant | Total |
|---|---|---|---|---|---|---|---|
| | n (%) | n/% | n/% | n/% | n/% | n/% | N |
| | Adults/ Parents | Adults/ Parents | Adults/ Parents | Adults/ Parents | Adults/ Parents | Adults/ Parents | Adults/ Parents |
| Have you been unsure whether you /your child belong to a particularly vulnerable group? | **353(45.9)** / **20(33.3)** | 164(21.3) / 13(21.7) | 88(11.4) / 11 (18.3) | 96 (11.7) / 7 (11.7) | 62 (8.0) / 8 (13.3) | 6 (0.7) / 1 (1.7) | 769 / 60 |
| To which extent do you feel you have control over you or your child becoming infected with COVID-19? | 45(5.8) / 11 (18.3) | 134(17.3) / **17(28.3)** | 243(31.4) / 16 (26.7) | **281(36.3)** / 12(20.0) | 72(9.3) / 4 (6.7) | 0 / 0 | 775 / 60 |
| Have you been unsure whether you should take different precautions than other people? | **299(39.4)** / **21(35.0)** | 189(24.9) / 15(25.0) | 90(11.9) / 10 (16.7) | 97(12.8) / 6 (10.0) | 83(10.9) / 8 (13.3) | 11(1.5) / 0 | 769 / 60 |
| Have you been unsure to what extent you/your child should isolate yourself? | **290(38.2)** / 16(26.7) | 239(31.4) / **18(30.0)** | 103(13.6) / 12 (20.0) | 85(11.2) / 6 (10.0) | 43(5.7) / 8 (13.3) | 9(1.2) / 0 | 769 / 60 |
| Have you been unsure about the use of protective equipment? | **593(77.7)** / **50(83.3)** | 101(13.2) / 8(13.3) | 39(5.11) / 1 (1.7) | 15(2.0) / 1 (1.7) | 15(2.0) / 0 | 6(0.8) / 0 | 769 / 60 |
| Have you been unsure how to deal with the risk of infection from your family? | **395(52.0)** / **25(41.7)** | 222(29.2) / 21(35.0) | 66(8.7) / 6 (10.0) | 49(6.4) / 3 (5.0) | 28(3.7) / 4 (6.7) | 9(1.2) / 1 (1.7) | 769 / 60 |
| Have you been unsure about what precautions your family should take to reduce the risk of infecting you? | **443(58.4)** / **39(65)** | 200(26.4) / 16(26.7) | 64(8.4) / 1(1.7) | 34(4.5) / 1 (1.7) | 18(2.4) / 3 (5.0) | 10(1.3) / 0 | 769 / 60 |
| Have you been unsure about what precautions to take when a person in your immediate family had to start school/education/ work after lockdown? | **338(52.6)** / 16(26.7) | 158(24.6) / **20(33.3)** | 70(10.9) / 9 (15.0) | 38(5.9) / 5 (8.3) | 38(5.9) / 10 (16.7 | 127(16.5) / 0 | 769 / 60 |

Numbers in bold illustrate the most frequent answer

(n = 258, 79.9%) and parents (n = 24, 72.7%) had moderate or greater confidence that their personal assistants did everything they could to prevent them or their child from getting infected with COVID-19 (moderate: 33 adults (10.2%) and 2 parents (6.1%), quite a bit: 98 adults (30.3%) and 11 parents (33.3%), extreme: 127 adults (39.3%) and 11 parents (33.3%)). Few had little confidence (n = 18 adults, 5.6% and n = 4 parents, 12.1%) or no confidence at all (n = 14 adults, 4.3% and n = 0 parents); 33 adults (10.2%) and 5 parents (15.2%) found the question irrelevant.

A proportion of patients (118 adults (36.5%) and 17 parents (51.5%)) stated that they had abstained from getting personal assistance for themselves or their child due to the risk of COVID-19 infection for a little of the time or more during the pandemic (little of the time: 46 adults (14.2%) and 8 parents (24.2%), some of the time: 40 adults (12.4%) and 2 parents (6.1%), quite a bit of the time: 18 adults (5.6%) and 3 parents (9.1%), most of the time: 14 adults (4.3%) and 4 parents (12.1%), not relevant: 34 adults (10.5%) and 7 parents (21.2%).

**3.2.6 COVID-19 infections and symptoms.** Of the 556 adults (71.9%) and 42 children (70%) that had been tested for COVID-19, 14 adults (1.7% of all adults) and one child (1.5% of all children) had tested positive. The mean age for this group was 53.65 years (range 11–88 years) and the diagnoses were limb girdle muscular dystrophy, heridiatry motor sensory neuropathy, Duchenne muscular dystrophy, myotonic dystrophy type 1, congenital myopathy, myastenia gravis, spinal muscular atrophy, and facioscapulohumeral dystrophy.

Reported COVID-19 symptoms were discomfort (n = 1, 6.7%), headache (n = 5, 33.3%), loss of smell (n = 4, 26.7%) or taste (n = 3, 20.0%), flu-like symptoms (n = 2, 13.3%), fatigue (n = 2, 13.3%), muscle ache (n = 1, 6.7%), sore throat (n = 1, 6.7%), fever (n = 1, 6.7%), gastrointestinal symptoms (n = 1, 6.7%), sensitivity to light and noise (n = 1, 6.7%), chest pain (n = 1, 6.7%) or cough (n = 1, 6.7%). Three adults (20.0%) did not experience any symptoms. None of those tested positive reported hospitalization or medical treatment.

**3.2.7 COVID-19 vaccinations.** 733/811 adults and 58/67 parents answered the questions on vaccine. Most adults (n = 629, 85.8%) and most parents (n = 45, 77.6%) would accept an invitation to receive a COVID-19 vaccine themselves. Fewer parents, albeit still the majority, wanted their child vaccinated (n = 40, 69.0%). 376 adults and 21 parents (n = 397) stated one or more reasons for accepting vaccinations. 325 of these (81.9%) said they wanted to get vaccinated for their own or their child's safety and 113 (28.5%) felt it was their societal duty to reduce the risk of infecting others.

82 adults (11.2%) and 9 parents (15.5%) were uncertain about whether to accept a vaccine and 15 parents (25.9%) were uncertain about getting their child vaccinated. Few did not want themselves (22 adults (3.0%) and 4 parents (6.9%)) or their child vaccinated (3 parents (5.2%)).

78 adults and 10 parents (n = 88) gave reasons for their uncertainty or refusal to get vaccinated. They sometimes gave more than one reason. These were concerns about side-effects or long-term effects (n = 53, 60.2%), insufficient information about the vaccine (n = 29, 33.0%), a feeling of not needing vaccinations e.g., due to good health, pregnancy or because they already had been infected with COVID-19 (n = 9, 10.2%).

## 3.3 Social health

**3.3.1 Work and education.** Questions related to the pandemic's influence on work life or education were answered by 487/503 adults and 64/67 parents (Table 5). Many adults had continued to work or study the same hours (33.1–40.2%) or experienced changes in work or study hours for only a little (25%) or some of the time (22.9%). Many adults had also been able to take the necessary precautions in their job/education (most of the time (8.4%), all of the time (28.7%)), but for some (21.2%) this was only possible some of the time or less. In general, adults (43.3%) or parents (53.1%) experienced understanding from work/education for most of the time or more during the pandemic and did not feel a pressure from family or employers or educational institutions in their decision to turn up physically or stay at home (43.8–68.6%). 162 adults (33.3%) and 13 parents (20.3%) had chosen to isolate from work/education some or all of the time.

Sixty-one parents out of 67 answered on behalf of their child (Table 5). Most children (n = 54, 88.5%) had spent less time in school or daycare during the pandemic. 8 children (13.1%) had been at home most of the time or the whole time during the pandemic. Most parents (n = 45, 73.8%) had not at any time felt pressure from school or daycare on their decisions to keep their child at home or at school/daycare, and more experienced this kind of support from their family (n = 55, 90.2%). 39 parents (63.9%) reported that they had been able to take the necessary precautions at school/daycare for most of the time or more, but 11 parents (18.0%) felt this was only possible a little of the time or none of the time.

**3.3.2 Leisure, social relations, and loneliness.** Questions on leisure activities were answered by 779/811 adults and by parents of 61/67 children. As a result of the pandemic, 81.7% of adults and 78.7% of children had reduced the time spent on leisure activities; 79% of adults and 73.8% of children had been limited in what kind of activities they had been able to engage in as a result of the pandemic and 77.5% of adults had spent less time on shopping (Table 5). 78.9% of adults and 62.3% of parents indicated that they had chosen to isolate themselves or their child from activities outside the home a little of the time or more due to the pandemic. Notably, a group of adults (40.5%) and children (29.5%) had been isolated from leisure activities outside their home for most or the whole time of the pandemic.

Both adults and children had experienced that socializing with family (67.8% of adults, 62.3% of children), friends (78% of adults, 63.9% of children) and others (75.36% of adults, 62.4% of children) has been moderately or more difficult due to the pandemic (Table 6).

**Table 5. Perceptions of how COVID-19 has influenced work/education, school, day-care and leisure time.**

| | None of the time | A little of the time | Some of the time | Most of the time | All of the time | Not relevant |
|---|---|---|---|---|---|---|
| | A/ Pa/C % | A/ Pa/C % | A/ Pa/C % | A/ Pa/C % | A/ Pa/C % | A/ Pa/C % |
| **Work and education /school-daycare** | | | | | | |
| Less time on work or education/school-daycare | **33.1**/18.8/11.5 | 11.9/15.6/19.7 | 13.1/**25.0/32.8** | 8/8/18.0 | 5.3/3.1/18.0 | 27.5/29.7/0 |
| More time on work or education/school-daycare | **40.2/35.9/91.8** | 12.5/17.2/0 | 10.4/7.8/3.3 | 5.5/5.7/0 | 2.3/3.1/1.7 | 29.2/31.3/3.3 |
| Accomplished less than I would like | **32.4**/14.1/- | 12.3/10.9/- | 13.8/**70.3**/- | 7.4/15.6/- | 3.5/57.8/- | 30.6/28.1/- |
| Limitations in what kind of work or educational activities I have been able to perform | **29.6**/20.3/- | 11.9/12.5/- | 14.0/**21.9**/- | 7.8/9.4/- | 7.0/4.7/- | 29.8/31.3/- |
| Difficulties in performing my work or educational activities (it took extra effort) | **34.3**/20.3/- | 13.6/18.8/- | 13.6/18.8/- | 12.9/7.8/- | 4.1/3.1/- | 29.8/31.3/- |
| Ability to take the precautions I felt were right for me/my child at work or education/school-daycare | 7.0/4.7/4.9 | 7.2/9.3/13.1 | 7.2/17.2/13.1 | 8.4/**29.7**/29.5 | **28.7**/12.5/**34.4** | 29.6/26.6/4.9 |
| Have met understanding for my situation as vulnerable due to my diagnosis or my child's diagnosis | 8.4/7.8/6.6 | 7.0/6.3/8.2 | 6.4/9.4/14.8 | 15.4/20.3/37.7 | **27.9/32.8/27.9** | 34.7/23.4/4.9 |
| Experienced pressure to stay at home (by employer or educational institution) | **50.1/51.6/73.8** | 5.1/1.6/11.5 | 4.1/3.1/8.2 | 1.6/6.3/1.6 | 1.2/4.7/1.6 | 37.8/32.8/3.3 |
| Experienced pressure to show up physically (by employer or educational institution) | **45.4/43.8/73.8** | 6.8/7.8/9.8 | 5.1/4.7/8.2 | 3.9/6.3/1.6 | 3.1/9.4/0 | 35.7/28.1/6.6 |
| Experienced pressure to stay at home (by my family) | **48.3/45.3/90.2** | 9.7/9.4/1.6 | 7.0/9.4/4.9 | 3.1/10.9/0 | 0.4/1.6/0 | 31.6/23.4/3.3 |
| Experienced pressure to show up physically (by my family) | **59.6/68.6/90.2** | 2.1/1.6/1.6 | 1.8/0/1.6 | 0.8/1.6/0 | 0.8/1.6/0 | 169/18/6.6 |
| I have isolated myself/my child physically from work/educational institution/school-daycare | **33.5/32.8/32.7** | 10.1/18.8/27.9 | 7.8/7.8/18.0 | 7.0/10.9/4.9 | 8.4/1.6/8.2 | 33.3/28.1/8.2 |
| **Leisure** | | | | | | |
| Less time on leisure activities | 18.3/-/6.6 | 10.9/-/18.0 | 16.8/-/11.5 | **28.2**/-/**27.9** | 25.8/-/21.3 | 10.0/-/14.8 |
| Limitation in what kind of leisure activities, I have been able to perform | 9.9/-/9.8 | 9.2/-/9.8 | 19.1/-/16.4 | **25.8**/-/**24.6** | 24.9/-/23.0 | 11.0/-/16.4 |
| Difficulties in performing leisure activities (it took extra effort) | **21.2**/-/**23.3** | 11.4/-/11.5 | 13.2/-/8.2 | 17.3/-/13.1 | 18.9/-/9.8 | 18.0/-/36.1 |
| I have spent less time shopping | 12.3/-/- | 10.8/-/- | 15.8/-/- | **25.5**/-/- | 25.4/-/- | 10.1/-/- |
| I have isolated myself/my child physically from leisure activities outside the house | 15.8/-/2.8 | 15.9/-/11.5 | 22.5/-/**21.3** | **26.1**/-/18.0 | 14.4/-/11.5 | 5.4/-/4.9 |

Questions on work/education/school were answered by 487 adults (A) 64 parents (Pa) and for 61 children (C). Questions on leisure activities were answered by 779/811 adults and parents of 61/67 children. Parents were not asked to rate their own leisure activities which is indicated by '-'. "Limitation in work" and "Shopping" was not rated for children and is similarly indicated with '-'. Note, for clarity, answers in this table are illustrated by percentages only for each subgroup. Numbers in bold illustrate the most frequent answer.

The median score on the UCLA loneliness scale for the adults was 6 (range 3–9). 273 adults (35.2% out of the 775 answering the questions) scored 7 or above on the UCLA loneliness scale, suggesting the experience of loneliness amongst these adults. Based on parental report, the median score on the UCLA loneliness scale for the children was 5.0 (range 3–9). 14 children (23.0% of those answering the questions) scored 7 or more on the UCLA loneliness scale, consistent with the experience of loneliness among these children.

**3.3.3 Stigmatization.** During the pandemic, most adults (63.6%) and many parents (46.7%) had experienced that others saw them or their child as more vulnerable than they did and many felt that their/their child's disability had become more apparent to themselves and others. Some adults (46%) and children (36.2%) had experienced social exclusion because others feared infecting them (Table 6).

**Table 6. Difficulties socializing and perception of stigmatization.**

| | Not at all n (%) | A little n (%) | Moderately n (%) | Quite a bit n (%) | Extremely n (%) | Not relevant n (%) | Total n |
|---|---|---|---|---|---|---|---|
| | Adults/ Parents | Adults/ Parents | Adults/Parents | Adults/ Parents | Adults/Parents | Adults/ Parents | Adults/ Parents |
| **Difficulties socializing with:** | | | | | | | |
| Family | 67(8.6) / 8 (13.1) | 176(22.6) / 14 (23.0) | 156(20.1) / **15 (24.6)** | **189(24.3)** / 11 (18.0) | 182(23.4) / 12 (19.7) | 8(1.0) / 1 (1.6) | 778 / 61 |
| Friends | 27(3.5) / 4 (6.6) | 123(15.8) / 13 (21.3) | 137(17.6) / 8 (13.1) | 19(24.6) / 15 (24.6) | **279(35.9)** / **16 (26.2)** | 21(2.7) / 5 (8.2) | 778 / 61 |
| Others | 39(5.0) / 2 (3.3) | 104(13.4) / **17 (27.8)** | 135(17.4) / 10 (16.4) | 188(24.16) / 14 (23.0) | **263(33.8)** / 14 (23.0) | 49(6.3) / 4 (6.6) | 778 / 61 |
| | None of the time | A little of the time | Some of the time | Most of the time | All of the time | Do not know | Total n |
| Other people see me/my child as more vulnerable than I do | 199(25.7) / 27 **(45.0)** | 149(19.3) / 18 (30.0) | **205(26.5)** / 6 (10.0) | 93(12.0) / 4 (14.3) | 45(5.8) / 0 | 82(10.6) / 5 (8.3) | 773 / 60 |
| Other people have excluded me/my child socially due to fear of infecting me/him/her | **330(42.7)** / 33 **(55)** | 147(19.0) / 10 (16.7) | 123(15.9) / 9 (15.0) | 62(8.0) / 3 (4.5) | 23(3.1) / 0 | 88(11.6) / 5 (8.3) | 773 / 60 |
| Other people are more concerned about my/my child's health than I am | 160(20.2) / 33 **(55.0)** | **207(26.1)** / 19 (31.7) | 175(22.9) / 3 (5.0) | 118(14.9) / 1 (1.5) | 44(5.6) / 0 | 88(11.1) / 4 (6.7) | 792 / 60 |
| My disability has become more apparent to myself/my child | **203(27.5)** / 22 **(36.7)** | 144(19.5) / 9 (15.0) | 143(29.4 / 14 (23.3) | 159(21.5) / 9 (15.0) | 50(6.8) / 4(6.7) | 40 (5.4) / 2 (3.3) | 739 / 60 |
| My disability has become more apparent to others | **203(27.1)** / 18 **(30.0)** | 162(21.5) / 17 (28.3) | 146(19.4) / 13 (21.7) | 122 (16.2) / 7 (11.6) | 50(6.6) / 0 | 69(9.2) / 5 (7.5) | 752 / 60 |

The table illustrates to what extent adults and children have experienced difficulties socializing with extended family and friends during the pandemic. Parents answered on behalf of the children. The table also shows adults' and parents' perception on how much of the time they have experienced other people seeing or treating them/their child as vulnerable during the pandemic. Numbers in bold illustrate the most frequent answer.

## 3.4 Mental health

**3.4.1 Anxiety.** Questions on anxiety and depression were filled in by 735/811 adults and 58/67 parents. The median score for anxiety on the HADS was 4 (range 0–19) for adults and 7 (range 0–19) for parents. 186 adults (25.3%) reported symptoms suggesting the presence of anxiety; 92 adults (12.5%) reported symptoms suggesting mild anxiety, 66 adults (9.0%) moderate anxiety and 28 adults (3.8%) severe anxiety. 549 adults (74.7%) scored within the normal range.

27 parents (46.6%) reported symptoms consistent with the presence of anxiety; 13 (22.4%) reported symptoms suggesting mild anxiety, 8 (13.8%) moderate anxiety and 6 (10.3%) severe anxiety. 31 parents (53.4%) scored within the normal range.

Anxiety related specifically to the pandemic including the CAS was scored by 734/811 adults and 58/67 parents. Median score on the CAS was 0 (range 0–20) in adults and 1 (range 0–17) in parents. 21 adults (2.9%) and 5 parents (8.6%) scored ≥ 9 on the CAS, suggesting the presence of COVID-19 related anxiety. 713 adults (97.1%) and 53 parents (91.4%) scored within the normal range.

298 adults (40.6%) and 31 parents (53.4%) reported that the pandemic had caused anxiety;196 adults (26.7%) and 20 parents (34.5%) felt slightly more anxious, 45 adults (6.1%) and 4 parents (6.9%) somewhat more anxious, 35 adults (4.8%) and 5 parents (8.6%) a lot more anxious, and 22 adults (3.0%) and 2 parents (3.4%) extremely more anxious due to the pandemic. 435 (59.3%) adults and 27 parents (46.6%) did not think the pandemic had affected their anxiety levels.

For children, 40 parents (69.0%) reported that their child was afraid of getting infected with COVID-19; 22 children (37.9%) were a little afraid, 6 (10.3%) moderately afraid, 8 (13.8%) quite a bit afraid and 4 (6.9%) extremely afraid. 16 parents (27.6%) reported that their child was not afraid. The parents of 2 children (3.4%) did not know whether their child was afraid.

**3.4.2 Depression.** The median score for depression on the HADS for adults was 3 (range 0–20) and for parents 4 (range 0–20). 150 adults (20.4%) and 16 parents (27.6%) reported symptoms on the HADS consistent with the presence of depression; 81 adults (11.0%) and 7 parents (12.1%) reported symptoms suggesting mild depression, 50 adults (6.8%) and 7 parents (12.1%) moderate depression, and 19 adults (2.6%) and 2 parents (3.4%) severe depression. 585 adults (79.6%) and 42 parents (72.4%) scored within the normal range.

Most adults (n = 479, 65.2%) and parents (n = 45, 77.6%) reported that the pandemic had caused a depressed mood: 313 adults (42.6%) and 27 parents (46.6%) reported a slightly more depressed mood as a consequence of the pandemic, 88 adults (12.0%) and 11 parents (19.0%) a moderately more depressed mood, 51 adults (6.9%) and 6 parents (10.3%) quite a bit more depressed mood, and 27 adults (3.7%) and 1 parent (1.7%) an extremely more depressed mood. 255 adults (34.7%) and 13 parents (22.4%) stated that the pandemic had not caused a depressed mood.

**3.4.3 Quality of life (QoL).** The influence of the pandemic on QoL was rated by 733/811 adults and for 58/67 children. Most adults (n = 582, 79.4%) and children (n = 47, 81%) experienced a reduced QOL due to the pandemic; 251 adults (34.2%) and 25 children (43.1%) experienced a slight decrease, 289 adults (39.4%) and 21 children (36.2%) a moderate decrease and 42 adults (5.7%) and 1 child (1.7%) an extreme decrease.

A small group of 41 adults (5.6%) and 5 children (8.6%) experienced improved QoL due to the pandemic with 22 adults (3%) and 2 children (3.4%) reporting a slight increase, 18 adults (2.5%) and 3 children (5.2%) a moderate increase and 1 adult (0.1%) an extreme increase. For 110 adults (15%) and 6 children (10.3%) their QoL was not affected by the pandemic.

**3.4.4 Correlations between perceived risk of severe illness if infected with COVID-19 and social isolation and psychological distress.** For adults, greater perceived risk of severe illness from COVID-19 was associated with greater loneliness ratings ($r_s$ = 0.341, p<0.001), lower QOL ($r_s$ = -0.248 p< 0.001), more time isolated from work/education ($r_s$ = 0.323, p<0.000) and leisure activities outside the house ($r_s$ = 0.389, p<0.001) and to a lesser extent with more symptoms of anxiety ($r_s$ = 0.166, p<0.001), depression ($r_s$ = 0.155, p<0.001), and corona-related anxiety ($r_s$ = 0.123, p<0.001).

Greater parental perceptions of children's risk of severe illness from COVID-19 was associated with greater duration of parental isolation from work ($r_s$ = 0.308, p = 0.017), greater parental anxiety ($r_s$ = 0.323, p = 0.013), greater parental coronavirus anxiety ($r_s$ = 0.405, p = 0.002), greater duration of isolation of the child from school or daycare ($r_s$ = 0.393, p = 0.002) and leisure activities ($r_s$ = 0.406, p = 0.001), and greater loneliness ratings of the child ($r_s$ = 0.592, p<0.001). It was also associated with children's fear of COVID-19 infection ($r_s$ = 0.507, p<0.001). Greater risk perceptions did not reach significance for associations with parental depression ($r_s$ = 0.249, p = 0.059) and the children's QOL ($r_s$ = -0.240, p = 0.069).

## 3.5 Positive consequences of the pandemic

When asked about positive consequences of the pandemic, 558/811 adults and 38/67 parents (n = 596) had provided comments and sometimes gave more than one reason. These were: more time with inhouse family (n = 189, 31.7%), less stress and more time for oneself (n = 113, 19.0%), better sleep, more energy, less pain (n = 57, 9.6%), more flexibility and efficiency at work/education, less transport time (n = 57, 9.6%), societal and environmental advantages

(n = 52, 8.7%), more time for immersing oneself and for enjoying nature (n = 51, 8.6%), fewer infections (n = 44, 7.4%). Some did not experience any positive consequences (n = 53, 8.9%) or did not know (n = 12, 2.0%).

## 4. Discussion

### 4.1 Discussion of results

To our knowledge this is the first study to investigate the pandemic's impact on biopsychosocial health, daily activities and QoL in a large population of children and adults with NMD. Our study population represents a broad variation of NMDs. Around 30% of the patients were non-ambulant and around 30% of the adults and 60% of the children experienced impaired cough. Thus, besides having NMD, the reduced coughing places many of the patients in the present study in a particularly vulnerable group for severe illness if infected with COVID-19, according to the definition of the Danish Health Authority. Our results should be seen in that light [6, 10].

**4.1.1 General health.** In relation to perception of health, our results show that a large proportion of adults with NMD experience a fair or poor general health which, for around 40% of adults, is somewhat worse or much worse than one year ago. Some 30% attributed these changes directly or indirectly to the pandemic. Parents generally reported their child's health as good and the same as a year ago. However, a fairly large group (20%) thought their child's health was worse. These results emphasize the negative impact of COVID-19 on health in patients with NMD.

**4.1.2 Healthcare, medical and hospital visits.** One of the explanations for a decline in health, may be that is has become more difficult for patients to take care of their health during the pandemic. Around two-thirds of both adults and parents found it more difficult to take care of their or their child's health during the pandemic and only half of adults and children could maintain their physiotherapy on a regular basis, which is consistent with other studies showing that physical activity in patients with NMD or other disabilities, has been reduced during the pandemic [17, 34]. Comments made by participants in the present study directly link indirect effects of the pandemic such as COVID-19 restrictions and changes in physiotherapy and treatment with poorer health, decreased physical function and poorer QOL.

Most planned medical and hospital visits for adults in our study were delivered as usual, or by video- or phone calls. Only a small group reported cancellations by their GP or themselves, whereas cancellations at hospitals were slightly more frequent, especially for children. These findings are consistent with another study that showed a negative impact on visits and treatments of NMD patients in hospital settings during lockdowns [7]. The use of telehealth is suggested for vulnerable groups if physical presence is not possible to ensure check-ups or access to GPs [3, 7]. However, this does not replace a physical examination, treatment, or exercise.

It is unfortunate that it has become more difficult for patients to take care of their health during the pandemic. Living with NMD implies a constant awareness to maintain physical functioning in order to postpone progression of the NMD as long as possible. Physiotherapy and regular hospital follow-ups are necessary for this [17, 18]. Our findings emphasize the importance of access to physiotherapy and hospital follow-ups in relation to further lockdowns or other pandemics.

**4.1.3 Perceived COVID-19 vulnerability, control, infections, symptoms, and vaccines.** Most of the adults perceived themselves to be at moderate or greater risk of severe illness if infected with COVID-19. Similar ratings were made for a little more than half of the children. Many in our study population reported breathing difficulties, and the ratings thus comply well with official risk evaluations [6, 10]. Many were not uncertain whether they or their child

belonged to a vulnerable group. This is interesting since risk groups are being discussed and revised repeatedly by health authorities [5, 11, 16].

Considering the high-risk ratings given by participants, it is surprising that many adults and parents reported that the reasons for accepting a COVID-19 vaccine was to protect others as much as themselves. One would expect the main reason for accepting a vaccine would be to protect themselves. In relation to this, many of the patients had been tested for COVID-19 which is in compliance with the general Danish strategy of frequent testing [2].

Interestingly, only 1.7% of adults and 1.5% children had been tested COVID-19 positive. Other reports have also been made of few NMD-patients with COVID-19 infections [12]. Patients with NMD in general only experienced mild flu-like symptoms, and none were hospitalized which suggests that they may not be at increased risk of severe illness in contrast with the official risk evaluations of severe illness from COVID-19 [6, 10]. The latter has primarily been made by extrapolation due to a lack of data on NMD-susceptibility to COVID-19. Nonetheless, we cannot exclude that patients who fell seriously ill might have opted out of answering the survey. Furthermore, patients may have been good at taking preventive measures to avoid infection.

Consistent with this, adults and parents were generally not uncertain about what precautions to take to avoid infections and most of the adults believed they had some extent of control over becoming infected with COVID-19. Moreover, most adults and parents had full confidence that personal assistants did whatever it took to prevent infection. Nonetheless, only half of the parents reported some degree of control over their child becoming infected which is not surprising as children may not be able to uphold restrictive measures in the same way as adults. Consistent with this, around half of the parents had opted out of getting personal assistance for their child from others than family for some of the time during the pandemic due to the risk of infection. This was the case for a third of the adults. To opt out of personal assistance and visits in the home may be explained as a way to obtain control—in this period of risk of COVID-19.

**4.1.4 Work, leisure, social relations, loneliness, and stigmatization.** The majority of adults and children had reduced time spent on leisure activities, but not work during the pandemic, and they reported to have isolated themselves or their child from activities outside the home during the pandemic. Notably, 13% of children had been isolated from school or daycare for most or the whole time of the pandemic and just under 30% of children had been isolated from leisure activities for most or the whole time. Spending time with family, friends and others was in general more difficult and the results indicated an experience of loneliness amongst some adults and children. These findings agree with other COVID-19 studies underlining the risk of social isolation and loneliness during a pandemic [35, 36]. The isolation, however, was not found to be one-sided from the patients with NMD in our study. Most adults and some parents had experienced, respectively, that they or their child was seen as more vulnerable by others resulting in social exclusion because others were afraid of infecting them with COVID-19 or were worried about the health of the patient with NMD. This may explain why some patients in the present study became more aware of their own disability. Another explanation for the increased awareness of their own or their child's disability may be that patients actively had to consider their risk of severe illness based on their NMD-diagnosis and related symptoms. Telehealth interventions have been suggested as a means to provide isolated populations with meaningful social contact [35].

**4.1.5 Mental health and quality of life.** A large proportion of adults and parents (respectively 25% and 46%) experienced symptoms consistent with anxiety. Around one fifth reported symptoms of depression. In general, the pandemic was seen to contribute to anxiety and a depressed mood. Several studies in the general population and among patients with chronic

disease showed somewhat similar findings highlighting the negative impact of COVID-19 on mental health (anxiety, distress, and depression) in many of their participants [37–39]. Additionally, a large proportion of patients in the present study reported decreased QoL during the pandemic. However, some reported an increase in QoL which may seem surprising, but this has also been shown in other groups of patients with chronic diseases during the pandemic [40]. Positive consequences of the pandemic were related to spending more time with family, having more energy due to less activities away from home, less stress, fewer infections and better work flexibility and efficiency.

**4.1.6 Consequences of perceived risk of severe illness if infected with COVID-19 for social isolation, psychological distress and QoL.** Interestingly, adults and parents who regarded themselves or their child to be at greater risk of severe illness were more likely to isolate themselves or their child from work or school and leisure activities and, for parents, to experience anxiety, in particular coronavirus anxiety. Associations between high-risk perception and greater anxiety and depression levels were also present for adults, albeit weak. Furthermore, adults who perceived themselves to be at high risk were more likely to experience loneliness and lower QoL. Children of parents who regarded them at high risk were also more likely to experience loneliness and to be scared of COVID-19. Thus, risk perceptions appear to have behavioral and psychological consequences and careful considerations to this should be made when categorizing risk groups. These consequences appear severe and out of proportion given our findings that few patients had actually fallen ill with COVID-19 and had only experienced mild symptoms. Nonetheless, causation cannot clearly be conferred as it is also possible that those who were more anxious or depressed or lonely were more likely to rate themselves or their child at severe risk due to a more worried and negative outlook on life.

## 4.2 Discussion of study limitations

Our sample of >25% of invited patients is considered acceptable in surveys. Our sample represents adults and children of all ages and a broad spectrum of NMD diagnoses and disabilities which vouch for the transferability to other contexts, countries, and populations of patients with NMD.

Due to data-protection and many small subgroups, we were unfortunately not able to look at subgroups of diagnoses and ages as this would have compromised anonymity. A limitation of the study is that we did not know the patients' health status before the pandemic. However, we tried to compensate for this by asking the patients what they perceived to have caused a change in their health and directly asking about the consequence of the pandemic for their mental health. The outcomes related to the children were reported by the parents which provided us with secondhand knowledge. In a future study it would be interesting to ask children directly about their perception of the impact of the pandemic. Nevertheless, our findings present novel and important information on the impact of the pandemic on biopsychosocial health and QoL.

## 5. Conclusions

Our results demonstrate novel knowledge on the COVID-19 pandemic's impact on biopsychosocial health and QoL of patients with NMD. The results show that physical health was affected with less access to hospital visits and treatment and especially physiotherapy. A minority of patients had suffered from COVID-19, experiencing none to mild flu-like symptoms. Socially, patients had spent less time on leisure activities, difficulties socializing with family and friends, and some experienced social exclusion and loneliness. Many patients reported symptoms consistent with anxiety and depression and perceived COVID-19 to cause such

negative feelings, and that their QoL moreover was negatively impacted by the pandemic. Especially patients perceived to be at high risk of severe illness experienced poor mental health, poor QoL and isolation.

The results emphasize that careful considerations should be taken before placing people into high-risk groups as well as the importance of professional counselling and support for vulnerable patients during a time of a pandemic, to avoid unnecessary isolation and risk of stigmatization. In addition, it is important to provide access to healthcare and physiotherapy to postpone progression of the NMD and maintain physical functioning. Specific information on NMD in relation to the pandemic, risk, symptoms, and vaccines is crucial. We believe our findings on NMDs are transferable to other contexts, countries, and chronic diseases.

## Acknowledgments

We would like to thank all the participants who generously shared their time by participating and filling out the survey.

## Author Contributions

**Conceptualization:** Charlotte Handberg, Ulla Werlauff, Ann-Lisbeth Højberg, Lone F. Knudsen.

**Data curation:** Charlotte Handberg, Ulla Werlauff, Ann-Lisbeth Højberg, Lone F. Knudsen.

**Formal analysis:** Charlotte Handberg, Ulla Werlauff, Ann-Lisbeth Højberg, Lone F. Knudsen.

**Investigation:** Charlotte Handberg, Ulla Werlauff, Ann-Lisbeth Højberg, Lone F. Knudsen.

**Methodology:** Charlotte Handberg, Ulla Werlauff, Ann-Lisbeth Højberg, Lone F. Knudsen.

**Project administration:** Charlotte Handberg, Ulla Werlauff, Lone F. Knudsen.

**Resources:** Ulla Werlauff.

**Software:** Lone F. Knudsen.

**Validation:** Charlotte Handberg, Ulla Werlauff, Ann-Lisbeth Højberg, Lone F. Knudsen.

**Visualization:** Charlotte Handberg, Ulla Werlauff, Ann-Lisbeth Højberg, Lone F. Knudsen.

**Writing – original draft:** Charlotte Handberg, Ulla Werlauff, Lone F. Knudsen.

**Writing – review & editing:** Charlotte Handberg, Ulla Werlauff, Ann-Lisbeth Højberg, Lone F. Knudsen.

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
