## [Decision Letter · Decision Letter 0]

12 May 2021

PONE-D-21-09727

Impact of the COVID-19 pandemic on biopsychosocial health and quality of life among Danish children and adults with neuromuscular diseases – Patient Reported Outcomes from a national survey

PLOS ONE

Dear Dr. Handberg,

Thank you for submitting your manuscript to PLOS ONE. After careful consideration, we feel that it has merit but does not fully meet PLOS ONE’s publication criteria as it currently stands. Therefore, we invite you to submit a revised version of the manuscript that addresses the points raised during the review process.

We look forward to receiving your revised manuscript.

Kind regards,

Gabriel A. Picone

Academic Editor

PLOS ONE

Journal Requirements:

3. Please provide additional details regarding participant consent.

In the ethics statement in the Methods and online submission information, please ensure that you have specified what type you obtained (for instance, written or verbal, and if verbal, how it was documented and witnessed).

If your study included minors, state whether you obtained consent from parents or guardians.

If the need for consent was waived by the ethics committee, please include this information.

4. In your Methods section, please provide a justification for the sample size used in your study, including any relevant power calculations (if applicable).

Furthermore, please include additional information regarding the survey or questionnaire used in the study and ensure that you have provided sufficient details that others could replicate the analyses. For instance, if you developed a questionnaire as part of this study and it is not under a copyright more restrictive than CC-BY, please include a copy, in both the original language and English, as Supporting Information.

6. Thank you for stating the following in the Title page of your manuscript:

'Funding

This work was supported by the National Rehabilitation Center for Neuromuscular Diseases.'

'The funders had no role in study design, data collection and analysis, decision to publish, or preparation of the manuscript.'

Please clarify the sources of funding (financial or material support) for your study. List the grants or organizations that supported your study, including funding received from your institution.State what role the funders took in the study. If the funders had no role in your study, please state: “The funders had no role in study design, data collection and analysis, decision to publish, or preparation of the manuscript.”If any authors received a salary from any of your funders, please state which authors and which funders.

*Please include your amended statements within your cover letter; we will change the online submission form on your behalf.*

Reviewers' comments:

Reviewer's Responses to Questions

**Comments to the Author**

1. Is the manuscript technically sound, and do the data support the conclusions?

Reviewer #1: Yes

2. Has the statistical analysis been performed appropriately and rigorously? 

Reviewer #1: Yes

3. Have the authors made all data underlying the findings in their manuscript fully available?

Reviewer #1: Yes

4. Is the manuscript presented in an intelligible fashion and written in standard English?

Reviewer #1: Yes

5. Review Comments to the Author

Reviewer #1: This article investigated the impact of the COVID-19 pandemic on biopsychosocial health, daily activities, and quality of life among children and adults with NMD, and assessed the prevalence of COVID-19 infection and the impact of this in NMD-patients.

This is a research article, correctly designed, with an updated bibliography. Moreover, in the introduction the authors could insert a recent paper, specific of the management of NMD patients affected by COVID-19, as Liguori S, Moretti A, Paoletta M, Gimigliano F, Iolascon G. Rehabilitation of Neuromuscular Diseases During COVID-19: Pitfalls and Opportunities. Front Neurol. 2021 Feb 19;12:626319. doi: 10.3389/fneur.2021.626319. PMID: 33679588; PMCID: PMC7933194.

However, although the results (section 3) are reported in a clear and exhaustive way, they are somewhat dispersed, therefore it is difficult to draw conclusions.

Furthermore, some points should be reviewed.

Specific comments

Title

Please report the abbreviation of neuromuscular diseases (NMD).

Abstract

Please report in full “neuromuscular diseases”.

2. Material and methods

Please describe the design of the study better.

3. Results

3.1. Study population

Please report what the missing data are.

In Table 1 “Occupational status” should be have less categories.

3.2 General health

3.2.1 Changes in general health

Please report the percentage of non-responders.

The term “most” is not correct for a percentage of 50.6.

3.2.2 Medical and hospital appointments

Word “had” is repeated, please delete one.

Physiotherapy

Please report the percentage of non-responders.

3.2.4 Perceived COVID-19 vulnerability and control

Please report the percentage of non-responders.

“Most adults and parents” should be quantified with a number.

3.2.5 Patients with need of personal assistance

Please report the percentage of non-responders.

3.2.6 COVID-19 infections and symptoms

Please report why “566 adults and 42 children had been tested for Covid-19”.

3.2.7 COVID-19 vaccinations

Please report the percentage of non-responders.

3.3 Social Health

3.3.1 Work and education

Please report the percentage of non-responders.

“Most adults” should be quantified with a number.

3.3.2 Leisure, social relations, and loneliness

Please report the percentage of non-responders.

“Majority” should be quantified with a number.

“Both adults and children had experienced that socializing with family, friend and others has been more difficult due to the pandemic” should be quantified with a number.

3.3.3 Stigmatization

The term “most” is not correct for a percentage of 46.7.

The term “some” should be quantified with a number.

3.4 Mental health

3.4.1 Anxiety

Please report the percentage of non-responders.

3.4.3 Quality of life (QoL)

Please report the percentage of non-responders.

3.4.4 Correlations between perceived risk of severe illness if infected with COVID-19 and social isolation and psychological distress

All values of “r” are too low to report conclusions; results of correlations are quite weak.

The term “moderately” is not correct for a value of “r” of 0.341.

3.5 Positive consequences of the pandemic

Please report the percentage of non-responders.

6. PLOS authors have the option to publish the peer review history of their article (what does this mean?). If published, this will include your full peer review and any attached files.

Reviewer #1: No

---

## [Author Response · Author response to Decision Letter 0]

4 Jun 2021

Dear Editorial board of PLOS ONE

Thank you for your forward interest in our manuscript. We are grateful that you decided to give us the chance to revise the manuscript according to the reviewer’s comments to achieve formal acceptance of the manuscript.

We hereby resubmit our revised manuscript entitled, Impact of the COVID-19 pandemic on biopsychosocial health and quality of life among Danish children and adults with neuromuscular diseases (NMD) – Patient Reported Outcomes from a national survey, by Charlotte Handberg, Ulla Werlauff, Ann-Lisbeth Højberg and Lone Knudsen.

We appreciate the useful review comments and have answered them below point by point and changed them accordingly in the ‘Revised manuscript with track changes’ (changes highlighted in red).

We sincerely hope that our changes are to your satisfaction.

On behalf of the authors, yours sincerely

Charlotte Handberg

Senior Researcher and Associate Professor, PhD

National Rehabilitation Center for Neuromuscular Diseases and Aarhus University, Denmark

New Funding statement: 

No external funding was received for this study.

New Data Availability Statement:

The data for this study contain potentially identifying patient information and we are therefore according to The Danish Data Protection Agency not able to share data publicly. The contact information for The Danish Data Protection Agency is: Address: Carl Jacobsens Vej 35, 2500 Valby, Denmark. Phone +45 33193200 mail: dt@datatilsynet.dk website: https://www.datatilsynet.dk/english  

Editor’s comments to author:

Answer to editor: 

The manuscript has been adjusted to meet the PLOS ONE’s style requirements.

2. Please review your reference list to ensure that it is complete and correct. If you have cited papers that have been retracted, please include the rationale for doing so in the manuscript text or remove these references and replace them with relevant current references. Any changes to the reference list should be mentioned in the rebuttal letter that accompanies your revised manuscript. If you need to cite a retracted article, indicate the article’s retracted status in the References list and also include a citation and full reference for the retraction notice.

Answer to editor:

The reference list has been reviewed to ensure it is complete and correct. 

3. Please provide additional details regarding participant consent.

In the ethics statement in the Methods and online submission information, please ensure that you have specified what type you obtained (for instance, written or verbal, and if verbal, how it was documented and witnessed). If your study included minors, state whether you obtained consent from parents or guardians. If the need for consent was waived by the ethics committee, please include this information.

Answer to editor:

Written consent was obtained directly from patients above 18 years of age and through the parents to patients under 18 years. This has been added to the ethics section. 

4. In your Methods section, please provide a justification for the sample size used in your study, including any relevant power calculations (if applicable). 

Furthermore, please include additional information regarding the survey or questionnaire used in the study and ensure that you have provided sufficient details that others could replicate the analyses. For instance, if you developed a questionnaire as part of this study and it is not under a copyright more restrictive than CC-BY, please include a copy, in both the original language and English, as Supporting Information.

Answer to editor:

As this was an investigative study of a new virus and its consequences, we decided to invite the whole NMD-population. We now state this in section 2.2 Setting and Sampling.

The questionnaire for our study included both self-developed questions as well as a number of standardized questionnaires, some of which are under copyright. We are thus not able to include a copy of the full questionnaire. We have, however, provided additional details of the questionnaire in the method section. If you would like us to send a copy of the self-developed questions, we are of course willing to provide those.

Answer to editor:

New Data Availability Statement which is also included in the cover letter:

The data for this study contain potentially identifying patient information and we are therefore according to The Danish Data Protection Agency not able to share data publicly. The contact information for The Danish Data Protection Agency is: Address: Carl Jacobsens Vej 35, 2500 Valby, Denmark. Phone +45 33193200 mail: dt@datatilsynet.dk website: https://www.datatilsynet.dk/english

6. Thank you for stating the following in the Title page of your manuscript:

'Funding

This work was supported by the National Rehabilitation Center for Neuromuscular Diseases.'

'The funders had no role in study design, data collection and analysis, decision to publish, or preparation of the manuscript.'

a. Please clarify the sources of funding (financial or material support) for your study. List the grants or organizations that supported your study, including funding received from your institution.

*Please include your amended statements within your cover letter; we will change the online submission form on your behalf.*

Answer to editor:

As suggested, we have deleted the funding-related text from the manuscript. We realize that our prior information on source of funding could be misinterpreted. We did not receive a grant to fund the study, hence no funders had influence on the study. New funding statement which is also included in the cover letter: ‘No external funding was received for this study.’

Reviewer Comments to Author:

1. Is the manuscript technically sound, and do the data support the conclusions?

Reviewer #1: Yes

2. Has the statistical analysis been performed appropriately and rigorously? 

Reviewer #1: Yes

3. Have the authors made all data underlying the findings in their manuscript fully available?

Reviewer #1: Yes

4. Is the manuscript presented in an intelligible fashion and written in standard English?

Reviewer #1: Yes

5. Review Comments to the Author

Reviewer #1: This article investigated the impact of the COVID-19 pandemic on biopsychosocial health, daily activities, and quality of life among children and adults with NMD, and assessed the prevalence of COVID-19 infection and the impact of this in NMD-patients.

This is a research article, correctly designed, with an updated bibliography. Moreover, in the introduction the authors could insert a recent paper, specific of the management of NMD patients affected by COVID-19, as Liguori S, Moretti A, Paoletta M, Gimigliano F, Iolascon G. Rehabilitation of Neuromuscular Diseases During COVID-19: Pitfalls and Opportunities. Front Neurol. 2021 Feb 19;12:626319. doi: 10.3389/fneur.2021.626319. PMID: 33679588; PMCID: PMC7933194.

However, although the results (section 3) are reported in a clear and exhaustive way, they are somewhat dispersed, therefore it is difficult to draw conclusions.

Furthermore, some points should be reviewed.

Answer to reviewer #1:

The suggested reference has been added to the introduction section.

The results section has been revised in relation to the comments below and changes has been made accordingly in the revised manuscript text and tables. 

With this revision we hope that the results are presented more distinctly and are easier to draw conclusion from.

Specific comments

Title

Please report the abbreviation of neuromuscular diseases (NMD).

Answer to reviewer #1:

The abbreviation has been added to the title.

Abstract

Please report in full “neuromuscular diseases”.

Answer to reviewer #1:

Full ‘neuromuscular diseases’ has been added to the abstract instead of the abbreviation.

2. Material and methods

Please describe the design of the study better.

Answer to reviewer #1:

We now describe the questionnaire in specific detail in the method section.

3. Results

3.1. Study population

Please report what the missing data are.

In Table 1 “Occupational status” should be have less categories.

Answer to reviewer #1:

Thank you to the reviewer for pointing out that we have not specified missing data and for the suggestion of less categories. Participants who had only filled in demographic information but left the rest of the questionnaire unanswered were excluded from the analysis. This is now explained in the Results section (3.1. Study population).

The categories for occupational status in Table 1 have been reduced from 12 to 8 categories.

3.2 General health

3.2.1 Changes in general health

Please report the percentage of non-responders. 

The term “most” is not correct for a percentage of 50.6. 

Answer to reviewer #1:

To clarify responders/non-responders; we have changed the wording to “804/811 adults” and “67/67 parents” (3.2.1 Changes in health).

Thank you for pointing out the mistake of using most when only half was the case. We have now changed the wording, so it reads: ‘When asked whether the general health had changed compared to one year ago, half of the adults (n=407, 50.6%) and most parents (n=45, 67.2%) rated their/their child’s health to be the same as a year ago.’

3.2.2 Medical and hospital appointments

Word “had” is repeated, please delete one.

Answer to reviewer #1:

Thank you for pointing this mistake out. The word “had” has been deleted.

Physiotherapy

Please report the percentage of non-responders.

Answer to reviewer #1:

To clarify responders/non-responders; we have changed the wording to “766/811 adults” and “67/67 parents” (3.2.3 Physiotherapy)

3.2.4 Perceived COVID-19 vulnerability and control

Please report the percentage of non-responders.

“Most adults and parents” should be quantified with a number.

Answer to reviewer #1:

To clarify responders/non-responders; we have changed the wording to “775/811 adults and 60/67 parents”

“Most adult and parents” refers to the numbers in Table 4; the percentage of adults and parents has now been added to the text.

3.2.5 Patients with need of personal assistance

Please report the percentage of non-responders

Answer to reviewer #1:

All adults and parents of children who received some form of personal assistance (full- or parttime at home, for leisure activities or/and at school/work) answered these questions. That is 323 out of 323 adults and 33 out of 33 parents of children. The text now reads:

‘323/323 adults and 33/33 parents of children with personal assistance (at home, for leisure activities, and/or at work/school/day-care) answered questions on concerns about infection from personal assistants.’

3.2.6 COVID-19 infections and symptoms

Please report why “566 adults and 42 children had been tested for Covid-19”.

Answer to reviewer #1:

In the method section under ‘COVID-19 infection and symptoms’ we now state the reason for a COVID-19 test. From an early start of the pandemic the Danish Health Authority recommended a COVID-19 test for persons with symptoms of the disease and persons who had been in contact with a person who had tested positive for COVID-19. 

3.2.7 COVID-19 vaccinations

Please report the percentage of non-responders.

Answer to reviewer #1:

To clarify responders/non-responders; we have changed the wording to “733/811 adults and 58/67 parents” (3.2.7 COVID-19 vaccinations).

3.3 Social Health

3.3.1 Work and education

Please report the percentage of non-responders.

“Most adults” should be quantified with a number.

Answer to reviewer #1:

To clarify responders/non-responders; we have changed the wording to “487/503 adults and 64/67 parents’ (3.3.1 work and education).

‘Most adults’ refers to Table 5. However, we now state numbers in text.

3.3.2 Leisure, social relations, and loneliness

Please report the percentage of non-responders.

“Majority” should be quantified with a number.

“Both adults and children had experienced that socializing with family, friend and others has been more difficult due to the pandemic” should be quantified with a number.

Answer to reviewer #1:

To clarify responders/non-responders; we have changed the wording to “779/811 adults” and “61/67 parents” (3.3.2 Leisure).

The wording majority has been corrected to the exact number: (3.3.2 Leisure).

The sentence “Both adults and children had experienced that socializing with family, friend and others has been more difficult due to the pandemic” refers to Table 6. However, the numbers have now been added in text so it reads:

Both adults and children had experienced that socializing with family (67.8% of adults, 62.3% of children), friends (78% of adults, 63.9% of children) and others (75.36% of adults, 62.4% of children) has been moderately or more difficult due to the pandemic (Table 6). 

3.3.3 Stigmatization

The term “most” is not correct for a percentage of 46.7.

The term “some” should be quantified with a number.

Answer to reviewer #1:

Thanks for pointing this out. The word ‘many’ has now been added before ‘parents’ so it now reads “most adults (63.6%) and many parents (46.7%)”

“Some” has been quantified with numbers. (3.3.3 stigmatization).

3.4 Mental health

3.4.1 Anxiety

Please report the percentage of non-responders.

Answer to reviewer #1:

To clarify responders/non-responders; we have changed the wording to “735/811 adults” and “58/67 parents” (3.4.1 Anxiety).

3.4.3 Quality of life (QoL)

Please report the percentage of non-responders.

Answer to reviewer #1:

To clarify responders/non-responders; we have changed the wording to “733/811 adults” and “58/67 parents” (3.4.3 Quality of life).

3.4.4 Correlations between perceived risk of severe illness if infected with COVID-19 and social isolation and psychological distress

All values of “r” are too low to report conclusions; results of correlations are quite weak.

The term “moderately” is not correct for a value of “r” of 0.341.

Answer to reviewer #1:

We thank the reviewer for this comment which has allowed us to reflect on these analyses. 

The value of Spearman’s r’s will always lie between -1 and +1. There are different recommendations regarding interpretation of the degree of association. Some consider an r-value between ±0.70 and ±0.90 a very strong correlation, and an r-value between ±0.40 and ±0.60 a strong correlation and an r-value between 0.30 and 0.39 a moderate correlation and an r-value of 0.19 to 0.29 a small or weak correlation. Others are more conservative and consider a value of 0.70 to 0.90 a strong correlation, a value of 0.40 to 0.69 a moderate correlation, 0.10-0.39 a weak correlation. See for instance Akoglu H. User’s guide to correlation coefficients. Turkish Journal of Emergency Medicine, 2018. 18: 91-93. However, arguments about the strength of the association can only be made when the p-value for the correlation analysis is significant which it was for all the r-values the reviewer has commented on. Graphing of the data was performed before all analyses were made to ensure a linear relationship.

Based on the reviewer’s comments, we have decided to take a somewhat more conservative approach and have deleted the word ‘moderately’. In addition, we have adjusted our assertions in the discussion section (4.1.6 Consequences of perceived risk of severe illness if infected with COVID-19 for social isolation, psychological distress and QoL).

3.5 Positive consequences of the pandemic

Please report the percentage of non-responders.

Answer to reviewer #1:

To clarify responders/non-responders; we have changed the wording to “558/811 adults and 38/67 parents (3.5 Positive consequences of the pandemic).

Thank you for your time and effort spend with our manuscript and all your useful comments. 

We sincerely hope that our changes are to your satisfaction.

---

## [Decision Letter · Decision Letter 1]

11 Jun 2021

Impact of the COVID-19 pandemic on biopsychosocial health and quality of life among Danish children and adults with neuromuscular diseases (NMD) – Patient Reported Outcomes from a national survey

PONE-D-21-09727R1

Dear Dr. Handberg,

We’re pleased to inform you that your manuscript has been judged scientifically suitable for publication and will be formally accepted for publication once it meets all outstanding technical requirements.

Kind regards,

Gabriel A. Picone

Academic Editor

PLOS ONE

Additional Editor Comments (optional):

Reviewers' comments:

Reviewer's Responses to Questions

**Comments to the Author**

1. If the authors have adequately addressed your comments raised in a previous round of review and you feel that this manuscript is now acceptable for publication, you may indicate that here to bypass the “Comments to the Author” section, enter your conflict of interest statement in the “Confidential to Editor” section, and submit your "Accept" recommendation.

Reviewer #1: All comments have been addressed

2. Is the manuscript technically sound, and do the data support the conclusions?

Reviewer #1: Yes

3. Has the statistical analysis been performed appropriately and rigorously? 

Reviewer #1: Yes

4. Have the authors made all data underlying the findings in their manuscript fully available?

Reviewer #1: Yes

5. Is the manuscript presented in an intelligible fashion and written in standard English?

Reviewer #1: Yes

6. Review Comments to the Author

Reviewer #1: The authors applied the corrections suggested. The paper results appropriate both in terms of methods and results. In my opinion, it's ready to be published.

7. PLOS authors have the option to publish the peer review history of their article (what does this mean?). If published, this will include your full peer review and any attached files.

Reviewer #1: No

---

## [Editor Report · Acceptance letter]

22 Jun 2021

PONE-D-21-09727R1 

Impact of the COVID-19 pandemic on biopsychosocial health and quality of life among Danish children and adults with neuromuscular diseases (NMD) – Patient Reported Outcomes from a national survey 

Dear Dr. Handberg:

I'm pleased to inform you that your manuscript has been deemed suitable for publication in PLOS ONE. Congratulations! Your manuscript is now with our production department. 

Kind regards, 

on behalf of

Dr. Gabriel A. Picone 

Academic Editor

PLOS ONE